# The Effect of Intratumoral Interrelation among FOXP3+ Regulatory T Cells on Treatment Response and Survival in Triple-Negative Breast Cancer

**DOI:** 10.3390/cancers14092138

**Published:** 2022-04-25

**Authors:** Noriko Goda, Chika Nakashima, Ichiro Nagamine, Sunao Otagaki

**Affiliations:** 1Department of Surgical Oncology, Research Institute for Radiation Biology and Medicine, Hiroshima University, 1-2-3 Kasumi, Minami-ku, Hiroshima 734-8551, Japan; 2Department of Surgery, Hiroshima Kyoritsu Hospital, 2-20-20 Nakasu, Asaminami-ku, Hiroshima 731-0121, Japan; chikam242012@gmail.com (C.N.); ichiron0214@outlook.jp (I.N.); sunaootagaki@gmail.com (S.O.)

**Keywords:** FOXP3, regulatory T cell, tumor-infiltrating lymphocyte, neoadjuvant chemotherapy, triple-negative breast cancer

## Abstract

**Simple Summary:**

Triple-negative breast cancer (TNBC) is a disease in which immunotherapy is more successful than in other subtypes of breast cancer. We describe a specific immune response involved in the infiltration of regulatory T cells in TNBC. Focusing on interleukin 33 (IL-33) and transforming growth factor beta 2 (TGFb2), which were identified by network analysis, we examined the therapeutic effect and prognosis of patients by immunohistostaining for these markers. We found that FOXP3 is a good prognostic factor for patients with high IL-33 and TGFb2 in the tumor. This finding may lead to the development of novel therapeutic strategies such as inducing these cytokines. It also provides deeper insight into the role of FOXP3, a universal marker of regulatory T cells, in TNBC.

**Abstract:**

Triple-negative breast cancer (TNBC) is characterized by an active immune response. We evaluated intratumoral interrelation between FOXP3+ tumor-infiltrating lymphocytes and other cytokines in TNBC. Network analysis refined cytokines significantly correlate with FOPX3 in TNBC. Information on the treatment response and prognosis of patients, and survival data from the TGCA and METABRIC databases were analyzed according to refined cytokines. Interleukin (IL)-33 was significantly expressed by TNBC cell lines compared to luminal cell lines (log_2_ fold change: 5.31, *p* < 0.001) and IL-33 and TGFB2 showed a strong correlation with FOXP3 in the TNBC cell line. Immunohistochemistry demonstrated that the IL-33 high group was a significant predictor of complete response of neoadjuvant chemotherapy (odds ratio (OR) 4.12, *p* < 0.05) and favorable survival compared to the IL-33 low group (OR 6.48, *p* < 0.05) in TNBC. Survival data from TGCA and METABRIC revealed that FOXP3 was a significantly favorable marker in the IL-33 high group compared to the low IL-33 low group (hazard ratio (HR) 2.1, *p* = 0.02), and the IL-33 high/TGFB2 high subgroup showed significant favorable prognosis in the FOXP3 high group compared to the FOPX3 low group in TNBC (HR 3.5, *p* = 0.01). IL-33 and TGFB2 were key cytokines of intratumoral interrelation among FOXP3 in TNBC.

## 1. Introduction

In 2021, approximately 2.2 million women were diagnosed with breast cancer, and nearly 700,000 women died of breast cancer [1]. Triple-negative breast cancer (TNBC), defined as nonexpression of estrogen receptor (ER) and progesterone receptor and no amplification or overexpression of human epidermal growth factor receptor 2 (HER2), accounts for 10–20% of breast cancers. The overall 5-year survival rate after a diagnosis of TNBC is 40%. The recurrence rate after surgery is as high as 25%, and the mortality rate within 3 months after relapse is 75% [2,3]. Chemotherapy remains the standard therapeutic approach for TNBC at all stages, and pathological complete response (pCR) after neoadjuvant chemotherapy is a favorable surrogate marker. In addition, post-neoadjuvant therapy to improve the prognosis of patients with non-pCR TNBC achieves a successful outcome [4]. Previous studies have shown that TNBC has high immunogenicity and tends to stimulate the production of a high level of tumor-infiltrating lymphocytes (TILs) [5,6]. TIL-rich or lymphocyte-predominant TNBC is associated with a good prognosis and response to chemotherapy [7,8]. This provides a strong rationale for the inhibition of tumors from escaping anticancer immune mechanisms, such as immune checkpoint inhibitors, including programmed death-ligand 1 (PD-L1) and cytotoxic T-lymphocyte-associated antigen 4 (CTLA-4), by immunotherapies in TNBC [9,10]. In addition, the evaluation of some cancer-related gene mutations, such as in *BRCA1/2*, is promising for obtaining additional molecular-targeted agent options [11,12]. Although some patients with TNBC are expected to have an improved prognosis, the prognosis of patients with total TNBC is yet to improve [13]. TILs comprise immunoprogressive or immunosuppressive components. Among TILs, regulatory T cells (Tregs), which express the transcription factor forkhead box P3 (FOXP3), are recognized by immunohistochemical (IHC) staining of FOXP3 positivity [14,15]. Previous studies have reported that the prognostic value of FOXP3+ TILs in breast cancer is worse because of their immunosuppressive function [16,17]. In contrast, some studies have reported that higher FOXP3+ TILs are associated with favorable outcomes [18,19]. The prognostic value of FOXP3+ TILs in breast cancer outcomes remains controversial [20].

In this study, we assessed the prognostic value of FOXP3+ TILs in TNBC from the standpoint of the immunological response network. We aimed to improve our understanding of the immunological relevance of FOXP3+ TILs in TNBC and develop a new immunotherapy strategy for TNBC.

## 2. Materials and Methods

### 2.1. Ethics Statement

The Institutional Review Board of Hiroshima Kyoritsu Hospital, Hiroshima, Japan, approved this study in accordance with the Good Clinical Practice guidelines and the Declaration of Helsinki (approval number: HK2020-3-01; approval date 1 March 2020. The requirement for informed consent was waived due to the retrospective nature of the study.

### 2.2. Patients and Tumor Samples

We selected formalin-fixed, paraffin-embedded (FFPE) tumor samples of patients with TNBC with a luminal subtype who underwent neoadjuvant chemotherapy and complete resection between December 2014 and March 2021 in the Department of Breast Surgery, Hiroshima Kyoritsu Hospital, Hiroshima, Japan. The neoadjuvant chemotherapy (NAC) regimen consisted of four cycles of docetaxel (75 mg/m^2^ every 3 weeks); 12 of paclitaxel (80 mg/m^2^ every week), followed by four of the FEC regimen (500 mg/m^2^ 5-fluorouracil, 100 mg/m^2^ epirubicin, and 500 mg/m^2^ cyclophosphamide every 3 weeks), four of the AC regimen (60 mg/m^2^ doxorubicin and 600 mg/m^2^ cyclophosphamide every 3 weeks), or four of the dose-dense AC regimen (60 mg/m^2^ doxorubicin and 600 mg/m^2^ cyclophosphamide every 2 weeks).

Male patients, patients with noninvasive or microinvasive carcinoma instead of primary breast cancer, and patients who could not receive the scheduled NAC were excluded from the study.

### 2.3. Pathological Assessment and Evaluation

Histological characteristics, including nuclear grade, ER and HER2 status, and the Ki-67 labeling index, were assessed using needle biopsy before NAC according to the American Society of Clinical Oncology/College of American Pathologists Guidelines, wherein the molecular subtypes of invasive breast cancer are classified as either triple-negative (ER− HER2−) or luminal (ER+ HER2−). The Ki-67 labeling index was scored according to the guidelines and tumors were classified as having high (≥20%) or low (<20%) proliferation potential [21]. Stromal TILs were assessed on hematoxylin and eosin-stained slides with the maximum number of tumor lesions, and lymphocyte-predominant breast cancer (LPBC) was defined as stromal TIL ≥ 50%. Nodal metastasis was assessed using fine-needle biopsy of lymph nodes before NAC. Pathological complete response (pCR) was defined as the absence of invasive residuals in the primary lesion and axillary lymph nodes.

### 2.4. Molecular Network and Pathway Analysis

The molecular network of differentially expressed genes was analyzed using KeyMolnet, an integrated platform for biological information (KM Data Co., Tokyo, Japan) [22].

The statistical significance of the concordance between the extracted network and canonical pathways was evaluated as “log_2_ fold change” using an algorithm defined in KeyMolnet. The “score” was calculated based on hypergeometric distribution between the searched molecular network and the canonical pathway.

In mRNA analysis, genes that differentially expressed mRNA in TNBC and luminal cell lines, calculated as log_2_ fold change ≤−5 and ≥5, were extracted (*p* ≤ 0.001). Meanwhile, in miRNA analysis, genes that differentially expressed miRNA, expressed as log_2_ fold change ≤−1 and ≥1, were extracted (*p* ≤ 0.001).

### 2.5. Immunohistochemistry

Breast cancer tissues were cut into serial 5 μm sections and transferred onto electrostatic slides for immunochemical analysis. These specimens were heated at 65 °C for 30 min, deparaffinized in xylene, and rehydrated in a graded ethanol series. Subsequently, endogenous peroxidase activity was blocked with 3% hydrogen peroxide for 10 min. Next, 5% bovine serum albumin/1× Tris-buffered saline and Tween-20 were used to reduce nonspecific background staining. IHC staining of FOXP3 (mouse, clone 236A/E7, ab20034; Abcam, Cambridge, UK), interleukin (IL)-33 (rabbit, ab207737; Abcam), and transforming growth factor beta 2 (TGFB2) (mouse, clone SD4, ab36495; Abcam) was performed according to the manufacturer’s instructions. The degree of positivity of the markers was graded as low, medium, or high by one pathologist by visual estimation, and by two surgeons according to the H score. H scores <100, 100–199, and ≥200 were graded as low, medium, and high, respectively.

### 2.6. External Gene Expression Data Analysis

In molecular network analysis, gene expression data were extracted from the Gene Expression Omnibus Database (https://www.ncbi.nim.nih.gov/geo (accessed on 20 January 2020)). mRNA expression analysis using RNA sequencing data of TNBC cell lines (MDA-MB-468, HCC70, and HCC 1143) and non-TNBC cell lines (MCF7, BT474, and T47D) was included. miRNA expression analysis data were extracted from microarray data of TNBC cell lines (MDA-MB-231 and CAL-51) and non-TNBC cell lines (ZR-75-1 and MCF7). In survival analysis, gene expression and survival data from TNBC samples were derived from The Cancer Genome Atlas (TGCA) [23] and the Molecular Taxonomy of Breast Cancer International Consortium (METABRIC) [24]. TNBC samples from patients who received chemotherapy were analyzed. Propensity score matching was performed between the groups according to other clinical data.

### 2.7. Statistical Analyses

Associations between groups and variables were evaluated using the Chi-squared test or Fisher’s exact test. The cutoff values for optimal gene expression were calculated using the maximization method of the log-rank *p*-value in the maxstat R package [25]. The survival information of TNBC patients and the mRNA levels of each case were extracted from the TGCA and METABRIC cohort databases and analyzed according to the level of refined cytokines. Survival curves using TCGA and METABRIC cohorts were calculated using the Kaplan–Meier method and compared using the log-rank test. Patient and tumor factors in each group were adjusted by propensity score matching, and values of *p*  < 0.05 were considered statistically significant.

## 3. Results

### 3.1. Molecular Network Analysis of Treg Infiltration of TNBC

The signaling pathways that correlated with *MMP*, *p53*, and *HIF* were significantly enriched, and transcriptional regulation of SMADs and miRNA were upstream in their pathways in TNBC cell lines compared with luminal cell lines (Figure 1a,b). Among the cytokines and chemokines significantly expressed in TNBC cell lines compared with luminal cell lines, only IL-33 was highly expressed (log_2_ fold change: 5.31732, *p* < 0.001) (Figure 1c).

Certain genes, including *MSN* (encodes moesin), *NT5E* (encodes CD73), *MMP2*, TGFB2, and IL-33, showed high scores in their correlation with FOXP3 (Score: *MSN*: 8.54, *NT5E*: 6.15, *MMP2*: 5.98, TGFB2: 5.73, and IL-33: 5.45) (Figure 1d), and notably, these genes were directly correlated with FOXP3 (Figure 2a)

The detailed network revealed that *MSN* was activated in the TGFB signaling pathway, *CD73* was regulated by *HIF1*, and *TGFB* was significantly activated by leucine aminopeptidase and *MMP2* (Figure 2b,c).

The levels of miR-200b, a positive regulator of IL-33 transcription, significantly decreased in TNBC cell lines compared with non-TNBC cell lines (log_2_ fold change: −0.65558, *p* < 0.001) (Figure 2d, left). The expression of caspase-1, which activates IL-33, was significantly high in TNBC cell lines compared with luminal cell lines (log_2_ fold change: 3.6958, *p* < 0.001) (Figure 2d, right). Among immune checkpoint-related molecules, *CD274* was significantly correlated with FOXP3 in TNBC cell lines compared with luminal cell lines (log_2_ fold change: 4.15994, *p* < 0.001) (Figure 1d). Thus, we focused on IL-33 and TGFB2 as key markers involved in FOXP3+ TIL infiltration in TNBC.

### 3.2. Histopathological Evaluation of Markers Involved in FOXP3 in TNBC

FOXP3, IL-33, and TGFB2 proteins were verified in the FFPE samples obtained by biopsy before NAC from TNBC and luminal subtype patients. Table 1 shows the patient characteristics following complete resection. The HER2 subtype, a typical subtype of breast cancer, was excluded to verify the pure network analysis results. FOXP3 was positive for 5–35% of the stromal TILs. IL-33 and TGFB2 were positive in the stromal tumor lesion (Figure 3a). Compared to the luminal subtype, TNBC showed a significantly high rate of LPBC (TNBC 35.0% vs. luminal 5.0%, *p* < 0.05) and high pCR rate to NAC (TNBC 45.0% vs. luminal 5.0%, *p* < 0.05) (Figure 3b, left). The TNBC group showed a higher ratio of positivity for FOXP3, IL-33, and TGFB2 than the luminal group (Figure 3b right). The pCR group significantly comprised more patients with LPBC than the non-pCR group (pCR 66.7% vs. non pCR 18.2%, *p* < 0.05) (Figure 3c, left). Moreover, the pCR group had a higher positivity for IL-33 and TGFB2 than the non-pCR group (medium + high ratio of IL-33: pCR 100% vs. non-pCR 55.6%, *p* < 0.05; medium + high ratio of TGFB2: pCR 77.8% vs. non-pCR 55.6%, *p* < 0.05) (Figure 3c, right). Logistic regression analysis of pCR revealed that LPBC and IL-33 were significant predictors of pCR (univariate analysis: LPBC, OR 2.13; IL-33, OR 3.5; all *p* < 0.05; multivariate analysis: LPBC, OR 5.45; IL-33, OR 4.12; all *p* < 0.05) in TNBC (Table 2). Logistic regression analysis of overall survival revealed LPBC and IL-33 were significant predictors of overall survival (univariate analysis: LPBC, OR 3.55; IL-33, OR 2.63; all *p* <0.05; multivariate analysis: LPBC, OR 8.45; IL-33, OR 6.48; all *p* < 0.05.) in TNBC (Table 3). The mean follow-up period of the study patients was 48 months.

### 3.3. Survival Analysis of TNBC According to the Level of FOXP3 mRNA

Next, we analyzed large public TNBC data from the TGCA and METABRIC databases. Regarding overall survival, FOXP3 was a significantly favorable marker in the IL-33 high group compared with the IL-33 low group (HR2.1 [1.5–4.3]; *p* = 0.02) (Figure 4b). The IL-33high/TGFB2high group showed the most significant favorable prognosis in patients with FOXP3-high TNBC (HR3.5 [2.6–6.3]; *p* = 0.01) (Figure 4e).

## 4. Discussion

We clarified the intratumoral interrelation between FOXP3+ regulatory T cells in TNBC and verified whether these findings were related to the therapeutic response to NAC and prognosis. We selected the luminal subtype instead of the HER2 subtype as a comparison group; this is because the HER2 subtype has a distinct tumor environment due to HER2-related pathways and an established treatment strategy (anti-HER2 agents). Our results showed that the subpopulations of FOXP3+ TILs were associated with tumoral inflammatory cytokine levels, treatment response to NAC, and prognosis in patients with TNBC. The prognostic value of FOXP3 depends on the cytokine levels in TNBC. We attempted to explore some cytokines specifically associated with FOXP3+ markers in TNBC using a refined network search. The prognostic value of Tregs in the context of IHC FOXP3 positivity or suppressive function of FOXP3+ TILs in breast cancer remains unclear. Yeong et al. reported that higher densities of intratumoral FOXP3+ TILs are associated with better prognosis in TNBC [26]. Some studies have reported that the balance of FOXP3+ TILs and other TILs is associated with breast cancer prognosis [27,28,29,30,31]; additionally, some studies have shown that FOXP3+ Tregs have heterogeneous subpopulations; “effector” Tregs (eTregs) have played roles in impeding antitumor immune responses, whereas “non-Tregs” play a role in antitumor immune responses [32,33,34]. These subpopulations cannot be detected by IHC staining because both are positive for FOXP3. Sugiyama et al. showed that FOXP3^high^ eTregs possess immunosuppressive functions, whereas FOXP3^low^ non-Tregs possess immunoprogressive functions and are associated with the production of inflammatory cytokines [34]. Saito et al. demonstrated that a subpopulation of FOXP3 TILs is associated with key cytokines, such as IL-12 and TBFB1, which influence the prognosis of colorectal cancer [35]. We focused on IL-33 and TGF-β as significant markers related to FOXP3+ TILs in TNBC. IL-33 is a cytokine belonging to the IL-1 family and is present in the nuclei of various cells, including epithelial cells. IL-33 and IL-1α are damage-associated proteins released extracellularly during tissue damage to transmit inflammatory signals [36,37]. The IL-33 signal is transmitted via the ST2 receptor. The IL-33 receptor is expressed in Tregs and is involved in proliferation and function [38]. The TGFB family supposedly inhibits cellular immune responses and is a potent inducer of the epithelial-to-mesenchymal transition in mammary cells. This transformation has been associated with the acquisition of tumor stem-like properties [39]. The TGFB family influences the cancer stem cell population; thus, cancer drug resistance may also be affected [40]. The diversity of the subpopulations of FOXP3+ TIL infiltration is related to these cytokines, and our findings may lead to new translational approaches for immunotherapy for TNBC, such as decreasing FOXP3^high^ eTregs or enhancing FOXP3^low^ non-Tregs by the injection of these cytokines. Our results also showed a significant relationship between Tregs and PD-L1 expression in TNBC.

This study has some limitations. Network analysis is a study on a mathematical model, and we have not conducted experiments to capture actual phenomena. We limited the candidate cytokines and data as well as the discussions about other markers. Only a limited number of samples underwent IHC. Our findings cannot elucidate the mechanism underlying the increase or decrease in the levels of IL-33 and other cytokines in the tumor microenvironment. Further validation studies and investigation of these findings may help develop new therapeutic targets for FOXP3+ TILs. Enhancement of tumoral IL-33 by injection or specific induction of a relevant antagonist may improve the treatment response and survival of patients with TNBC. Another translational strategy to identify a more effective group for PD-L1 inhibitors according to FOXP3+ TIL levels in TNBC may be effective.

## 5. Conclusions

Our findings indicate that IL-33 and TGFB2 were key cytokines of intratumoral interrelation among FOXP3 in TNBC. FOXP3+ TIL infiltration is associated with tumoral expression of IL-33, and the prognostic value of FOXP3 depends on active expression of these cytokines in TNBC.

## Figures and Tables

**Figure 1 cancers-14-02138-f001:**
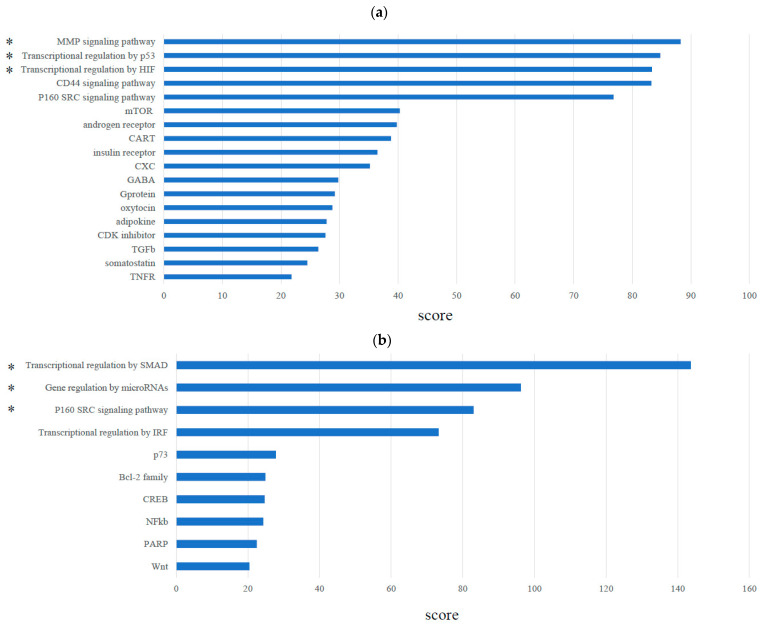
Representative molecular and genetic interrelation between triple-negative breast cancer (TNBC) cell lines (MDA-MB-231 and CAL-51) and luminal cell lines (ZR-75-1 and MCF7). (**a**) Enriched signaling pathways, (**b**) transcriptional regulation of upstream pathway, (**c**) cytokines and chemokine, and (**d**) genes significantly correlated with forkhead box P3 (FOXP3) in TNBC cell lines compared with the luminal cell lines. Each log_2_ fold change value was ranked. The score was calculated based on hypergeometric distribution between the searched molecular network and the canonical pathway. * indicates significant pathway in TNBC compared to luminal subtype.

**Figure 2 cancers-14-02138-f002:**
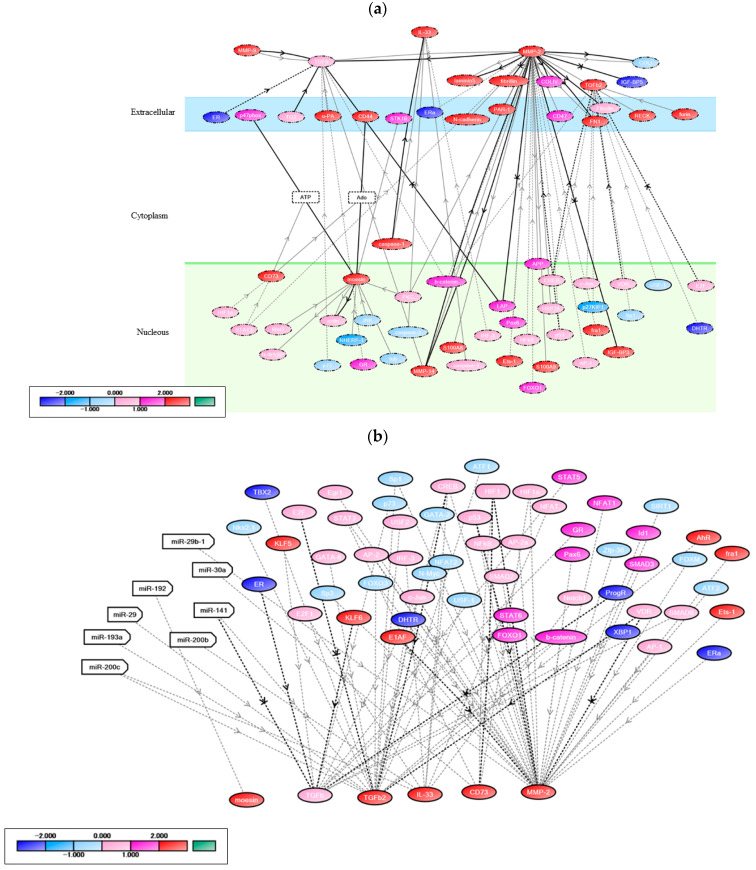
Relationship diagram of genes correlated with FOXP3 in TNBC revealed by network analysis. The solid lines indicate direct regulation, and the dotted lines indirect regulation. Color gradations of ovals indicate the expression level of each gene (left down). (**a**) Specific genes correlated with FOXP3 in TNBC cell lines (MDA-MB-231 and CAL-51). (**b**) Refined regulation relationship of *moesin*, transforming growth factor beta 2 (TGFB2)*,* interleukin (*IL*)*-33*, *CD73*, and *MMP2*. (**c**) Refined regulation relationship of *TGFB*. (**d**) Refined regulation relationship of IL-33.

**Figure 3 cancers-14-02138-f003:**
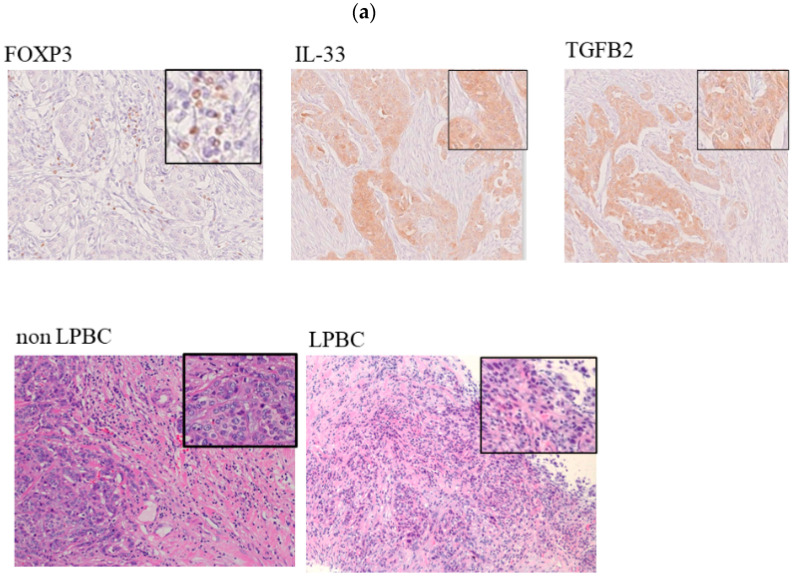
Histopathological assessment of FOXP3+ tumor-infiltrating lymphocytes (TILs) in TNBC and luminal subtype biopsy samples of patients who underwent neoadjuvant chemotherapy (NAC). All samples (TNBC, *n* = 20; luminal, *n* = 20) were obtained by biopsy of tumors before NAC. (**a**) Immunohistochemical staining of FOXP3, IL-33, and TGFB2 markers, and (above) non-lymphocyte-predominant breast cancer (LPBC) (stromal TILs < 50%) and (below) LPBC (stromal TILs ≥ 50%). Representative ×100 magnification and ×400 magnification (top right) images of each case. (**b**) Details about the incidence of stromal TILs (non-LPBC/LPBC), treatment response to NAC (non-pathological complete response (pCR) and pCR), FOXP3, IL-33, and TGFB2 in TNBC and luminal subtype. (**c**) Details about the incidence of stromal TILs and positivity of FOXP3, IL-33, and TGFB2 in TNBC. The degree of positivity of markers was determined in the field of view of the maximum fracture surface of the tumor. FOXP3: <3%, low; 3–10%, medium; and ≤10%, high of total TILs. IL-33 and TGFB2: <5%, low; 5–30%, medium; and ≤30%, high of total tumor.

**Figure 4 cancers-14-02138-f004:**
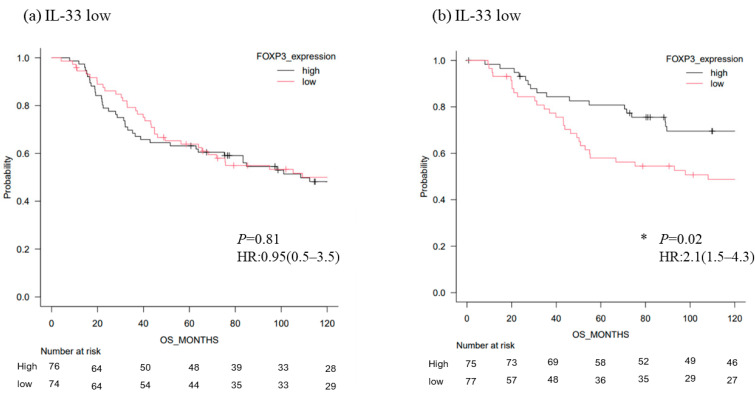
Overall survival of patients with TNBC in TGCA and METABRIC cohorts according to FOXP3 mRNA levels under different conditions of IL-33 and TGFB2 mRNA levels (*n* = 302). (**a**) FOXP3 high vs. low in the IL-33-low group (*n* = 200). (**b**) FOXP3 high vs. low in the IL-33-high group (*n* = 102). (**c**) FOXP3 high vs. low in the TGFB2-low group (*n* = 153). (**d**) FOXP3 high vs. low in the TGFB2-high group (*n* = 153). (**e**) Four groups (IL-33 high/TGFB2 high, IL-33 high/TGFB2 low, IL-33 low/TGFB2 high, and IL-33 low/TGFB2 low groups) in the FOXP3-high group (*n* = 135). (**f**) Four groups (IL-33 high/TGFB2 high, IL-33 high/TGFB2 low, IL-33 low/TGFB2 high, and IL-33 low/TGFB2 low groups) in the FOXP3-low group (*n* = 167). *: *p* < 0.05.

**Table 1 cancers-14-02138-t001:** Patient characteristics (pre NAC information).

Variate	TNBC (*n* = 20) Number (%)	Luminal (*n* = 20) Number (%)
Age (year), median (range)	57 (38–65)	59 (40–66)
Histological type		
Infiltrating duct carcinoma	16 (80)	15 (75)
Lobular carcinoma	0 (0)	4 (20)
Other	4 (20)	1 (5)
T status		
T1	3 (15)	0 (0)
T2	7 (35)	12 (60)
T3	6 (30)	5 (25)
T4	4 (20)	3 (15)
Nodal metastasis		
Negative	9 (45)	3 (15)
Positive	11 (55)	17 (85)
Histological grade		
1	1 (5)	2 (10)
2	8 (40)	9 (45)
3	11 (55)	9 (45)
LVI		
Negative	5 (25)	12 (60)
Positive	15 (75)	8 (40)
Ki-67 labeling index		
<20%	2 (10)	6 (30)
≥20%	18 (90)	14 (70)
Stromal TILs		
Non-LPBC (stromal TIL <50%)	13 (65)	19 (95)
LPBC (stromal TILs ≥50%)	7 (35)	1 (5)
Neoadjuvant chemotherapy		
Non-pCR	11 (55)	19 (95)
pCR	9 (45)	1 (5)

ER, estrogen receptor; HER2, human epidermal growth factor receptor 2; LPBC, lymphocyte-predominant breast cancer; LVI, lymphovascular invasion; pCR, pathological complete response; TILs, tumor-infiltrating lymphocytes.

**Table 2 cancers-14-02138-t002:** Univariate and multivariate analyses for predicting pathological complete response in TNBC.

Variate	Univariate Analysis		Multivariate Analysis	
	OR (95% CI)	*p*-Value	OR (95% CI)	*p*-Value
Ki-67 labeling index (≥20% vs. <20%)	2.34 (0.48–9.79)	0.158	1.89 (0.33–8.97)	0.264
TILs (LPBC vs. non LPBC)	2.13 (1.88–5.648)	0.002	5.45 (1.08–13.8)	0.002
Histological grade (0–1 vs. 2–3)	1.89 (0.22–5.98)	0.685	2.17 (0.45–6.58)	0.475
LVI (positive vs. negative)	5.12 (0.68–6.85)	0.445	4.98 (0.59–1.57)	0.555
IL33 (high vs. low)	3.54 (1.65–6.948)	0.001	6.32 (2.65–12.5)	0.003
TGFb2 (high vs. low)	4.12 (0.95–2.32)	0.542	3.85 (0.54–3.98)	0.335

CI, confidence interval; OR, odds ratio; LPBC, lymphocyte predominant breast cancer.

**Table 3 cancers-14-02138-t003:** Univariate and multivariate analyses for overall survival in TNBC.

Variate	Univariate Analysis		Multivariate Analysis	
	HR (95% CI)	*p*-Value	HR (95% CI)	*p*-Value
Ki-67 labeling index (≥20% vs. <20%)	3.25 (0.84–5.23)	0.658	2.45 (0.33–9.65)	0.678
TILs (LPBC vs. non LPBC)	3.42 (2.33–8.65)	0.008	6.54 (1.45–21.4)	0.015
Histological grade (0–1 vs. 2–3)	0.54 (0.24–5.75)	0.651	0.98 (0.35–5.42)	0.524
LVI (positive vs. negative)	2.57 (0.65–6.51)	0.447	5.26 (1.54–15.4)	0.284
Treatment response of NAC (pCR vs. non pCR)	3.55 (2.04–9.58)	0.002	8.45 (1.98–14.5)	0.001
IL33 (high vs. low)	4.12 (1.65–12.65)	0.003	6.48 (1.59–17.5)	0.004
TGFb2 (high vs. low)	2.31 (0.29–5.75)	0.514	1.57 (0.33–9.87)	0.246

CI, confidence interval; HR, hazard ratio; LPBC, lymphocyte predominant breast cancer; LVI, lymphovascular invasion; TILs, tumor-infiltrating lymphocytes.

## Data Availability

The datasets used and analyzed during the current study are available from the corresponding author on reasonable request.

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
