# Peer review of "The Effect of Intratumoral Interrelation among FOXP3+ Regulatory T Cells on Treatment Response and Survival in Triple-Negative Breast Cancer"

_cancers, 2022, doi:10.3390/cancers14092138_

Round 1
Reviewer 1 Report
- In the abstract setiion spelling of "imformation" was wrong
- Please explain why the 20% was chosen as the cut-off?
- And if the histology gradeing of the lobular carcinoma the same as IDC?
- In table 1 shows the T status of the tumro pre NAC, how dose the tumor size be measured ?
Author Response
Response to Reviewer 1
・In the abstract section spelling of "imformation" was wrong.
Response 1: Thank you for pointing out the errors. I corrected the spelling of "information".
・Please explain why the 20% was chosen as the cut-off?
Response 2: The Ki-67 labeling index is a generally accepted indicator of malignancy (proliferation potential) in breast cancer research. The cut-off value of Ki-67 labeling index is also generally around 20%. We set the cut-off value to 20% based on the prevailing norms.
・And if the histology grading of the lobular carcinoma the same as IDC?
Response 3: The histological grading of the lobular carcinoma is the same as that of IDC.
・In Table 1 shows the T status of the tumor pre-NAC, how does the tumor size be measured?
Response 4: Thank you for your inquiry about T stage in Table 1. The T stage was determined using the measurements obtained by imaging examination of each patient.
Reviewer 2 Report
Goda et al. assessed the correlation between FOXP3 expressing T-cells and cytokine expression with the prognostic outlook of triple negative breast cancer. In recent years, immunotherapy using immune checkpoint inhibitors has shown considerable promise for the management of triple-negative breast cancer, which is known to have an aggressive outlook. A detailed analysis of the immune microenvironment, more specifically, the role of the regulator T-cells is crucial as this can provide valuable information for refining patient management strategies.
For this purpose, the authors utilized immunohistochemistry (IHC) with FOXP3, IL33, and TGFB2, and also analysed publicly available dataset containing RNA-sequencing performed on cell-lines. The authors observed that tumors having a higher proportion of FOXP3 cells and higher levels of TGFB2 and IL33 expression associate with a better prognosis.
While the research question pursued by this study is highly relevant and the authors do uncover some interesting findings, there are several crucial methodological drawbacks of this study.
- The authors intended to study a specific subpopulation of T-cells, i.e., FOXP3 expressing T regulatory cells, and utilized IHC for this purpose. It is unclear to me why the authors did not include additional markers in their panel, e.g., CD3, CD4, CD8, to better characterize the (sub)populations of T-cells. The authors could have also utilized CD45RO / CD45RA to identify naiive and memory T-cell.
- This study majorly derives the conclusion on the analysis of immunohistochemical expression. Despite that, the article provides a very superficial description of the IHC assessment. There is some missing relevant information in this respect, which the authors could consider adding to their article. Was all IHC assessment performed by visual estimation? Or was the analysis performed using some digital image analysis software, e.g., QuPath / Image J? What scoring system did the authors utilize? Did they use an H-score? These details are not only vital for judging the quality of the manuscript, but also for future validation studies on this aspect.
- The authors could discuss what additional techniques could be used to better refine the levels of IL-33 for predicting the prognosis of the patients.
- The sample size in this study was quite small, and the authors should mention this as a limitation in the Discussion. The authors could also state that further validation studies will be required.
- In the section 2.7 Statistical analyses, the authors could mention the exact survival parameters that were measured.
- Did the authors also study the expression of FOXP3, IL-33, and TGFB2 in ductal adenocarcinoma situ – the non-obligate precursor of triple-negative breast cancer? Were there any specific patterns / differences in the expression of these markers in invasive breast carcinoma and ductal adenocarcinoma in-situ?
- There is a typo in section 2.1 Ethics statement under Materials and Methods. The year 2020 has been mistyped as 20200.
Author Response
1: The authors intended to study a specific subpopulation of T-cells, i.e., FOXP3 expressing 1: T regulatory cells, and utilized IHC for this purpose. It is unclear to me why the authors did not include additional markers in their panel, e.g., CD3, CD4, CD8, to better characterize the (sub)populations of T-cells. The authors could have also utilized CD45RO / CD45RA to identify naïve and memory T-cells.
Response 1: Thank you for your comment. We focused on FOXP3 because the study aimed to acquire new insights into FOXP3, a universal marker of regulatory T cells. We think it is understandable that not using many markers is considered to be a limitation, but we request that you focus on the important results arising from our network analysis.
2: This study majorly derives the conclusion on the analysis of immunohistochemical expression. Despite that, the article provides a very superficial description of the IHC assessment. There is some missing relevant information in this respect, which the authors could consider adding to their article. Was all IHC assessment performed by visual estimation? Or was the analysis performed using some digital image analysis software, e.g., QuPath / Image J? What scoring system did the authors utilize? Did they use an H-score? These details are not only vital for judging the quality of the manuscript, but also for future validation studies on this aspect.
Response 2: Thank you for pointing out this missing information. The IHC assessment was performed and validated by a pathologist and two surgeons and an H-score was used. We have added this information to Section 2.5 (Immunohistochemistry).
3: The authors could discuss what additional techniques could be used to better refine the levels of IL-33 for predicting the prognosis of the patients.
Response 3: Thank you for the suggestion. We added some wording to the last paragraph of the Discussion section to mention techniques that could be used to refine the levels of IL-33 in TNBC patients.
4: The sample size in this study was quite small, and the authors should mention this as a limitation in the Discussion. The authors could also state that further validation studies will be required.
Response 4: Thank you for these suggestions. We added the small sample size of IHC as a limitation and stated that further validation studies are required in the last paragraph of the Discussion section.
5: In the section 2.7 Statistical analyses, the authors could mention the exact survival parameters that were measured.
Response 5: Thank you for pointing out this missing information. We added information about data extraction and the parameters that were measured, to Section 2.7 (Statistical analyses).
6: Did the authors also study the expression of FOXP3, IL-33, and TGFB2 in ductal adenocarcinoma in situ – the non-obligate precursor of triple-negative breast cancer? Were there any specific patterns / differences in the expression of these markers in invasive breast carcinoma and ductal adenocarcinoma in-situ?
Response 6: Thank you for the question. We did not consider ductal adenocarcinoma in situ in our study (described in Section 2.2. “Patients and tumor samples"). It would be interesting to investigate the difference in expression of these markers between invasive breast cancer and non-invasive breast cancer. We will consider investigating this in our future research.
7: There is a typo in section 2.1 Ethics statement under Materials and Methods. The year 2020 has been mistyped as 20200.
Response 7: Thank you for pointing out this error. I corrected the year to "2020" in Section 2.1 (Ethics statement) under Materials and Methods.
Reviewer 3 Report
The manuscript The effect of intratumoral interrelation... is a straightforward, relatively narrowly focused text based on mathematical analysis of data. The authors themselves declare some limitation of the study, which is appreciable.
The study brings interesting data on triple negative breast cancers, which represents quite significant therapeutic problem. The results may support including of immunotherapeutic protocols in the treatment. Indeed, a poorly characterized population of FoxP3+ cells plays a critical role in the tuning of cytokine milieu and responses to neoadjuvant therapy. For the readers it would be helpful, if the relevant text in the Introduction on the FoxP3+ population, either suppressive or without the suppressive effect, would be somewhat extended.
In the Fig.2, namely a) and b), it is rather difficult to interpret the gene designations, even on the computer display. An enhancement of the graphic resolution is advisable.
In Fig. 1c) 1d), in the last place on the bottom, there are two columns always the same: is that correct?
Typing error in Abstract: com-pared
Author Response
1: In Fig.2, namely a) and b), it is rather difficult to interpret the gene designations, even on the computer display. An enhancement of the graphic resolution is advisable.
Response 1: Thank you for pointing this out. We have reformatted Fig. 2 to try to enhance the resolution and clarity.
2: In Fig. 1c) 1d), in the last place on the bottom, there are two columns always the same: is that correct?
Response 2: Thank you for pointing out the duplication of the results. We revised Fig.1c and 1d.
3: Typing error in Abstract: com-pared
Response 3: Thank you very much for pointing out the error. We revised "compared" in abstract and fixed the formatting bug in the template that was causing words to split at the end of lines.
Reviewer 4 Report
The manuscript titled: “The effect of intratumoral interrelation among FOXP3+ regulatory T cells on treatment response and survival in triple-negative breast cancer” is well written. However, Data needs to be refined in a presentable manner, and minor points need to be addressed before acceptance.
Minor comment:
- Figures in the manuscript are lack clarity. I request the authors to re-adjust the size of the figures. Text in the figures also needs to be refined. Increase the quality of images.
Author Response
Minor comment:
- Figures in the manuscript are lack clarity. I request the authors to re-adjust the size of the figures. Text in the figures also needs to be refined. Increase the quality of images.
Response 1: Thank you for pointing this out. We have reformatted all the figures to improve their clarity as much as possible.
Round 2
Reviewer 1 Report
I have no further comments about the manuscript
Reviewer 2 Report
No further comments.